# T-Cadherin Deficiency Is Associated with Increased Blood Pressure after Physical Activity

**DOI:** 10.3390/ijms241814204

**Published:** 2023-09-17

**Authors:** Vladimir S. Popov, Ilya B. Brodsky, Maria N. Balatskaya, Alexander V. Balatskiy, Ilia D. Ozhimalov, Maria A. Kulebyakina, Ekaterina V. Semina, Mikhail S. Arbatskiy, Viktoria S. Isakova, Polina S. Klimovich, Veronika Y. Sysoeva, Natalia I. Kalinina, Vsevolod A. Tkachuk, Kseniya A. Rubina

**Affiliations:** 1Faculty of Medicine, Lomonosov Moscow State University, Lomonosovsky Ave., 27/1, 119991 Moscow, Russiabrodskiy-i@yandex.ru (I.B.B.); m.balatskaya@gmail.com (M.N.B.);; 2V.I. Kulakov National Medical Center of Obstetrics Gynecology and Perinatology, Akademika Oparina Street, 4, 117198 Moscow, Russia

**Keywords:** T-cadherin, *Cdh13*, adiponectin, treadmill, blood pressure, AMPK

## Abstract

T-cadherin is a regulator of blood vessel remodeling and angiogenesis, involved in adiponectin-mediated protective effects in the cardiovascular system and in skeletal muscles. GWAS study has previously demonstrated a SNP in the *Cdh13* gene to be associated with hypertension. However, the role of T-cadherin in regulating blood pressure has not been experimentally elucidated. Herein, we generated Cdh13^∆Exon3^ mice lacking exon 3 in the *Cdh13* gene and described their phenotype. Cdh13^∆Exon3^ mice exhibited normal gross morphology, life expectancy, and breeding capacity. Meanwhile, their body weight was considerably lower than of WT mice. When running on a treadmill, the time spent running and the distance covered by Cdh13^∆Exon3^ mice was similar to that of WT. The resting blood pressure in Cdh13^∆Exon3^ mice was slightly higher than in WT, however, upon intensive physical training their systolic blood pressure was significantly elevated. While adiponectin content in the myocardium of Cdh13^∆Exon3^ and WT mice was within the same range, adiponectin plasma level was 4.37-fold higher in Cdh13^∆Exon3^ mice. Moreover, intensive physical training augmented the AMPK phosphorylation in the skeletal muscles and myocardium of Cdh13^∆Exon3^ mice as compared to WT. Our data highlight a critically important role of T-cadherin in regulation of blood pressure and stamina in mice, and may shed light on the pathogenesis of hypertension.

## 1. Introduction

T-cadherin is a member of the cadherin family, which are molecules of cell adhesion required for proper cellular function and tissue architecture maintenance. However, T-cadherin is a non-canonical representative of cadherins. While T-cadherin’s extracellular part consists of five Ca^2+^-binding domains typical of classical cadherins, it lacks the characteristic of its transmembrane and cytoplasmic parts being anchored to the plasma membrane via a glycosylphosphatidylinositol (GPI) [1]. Besides the homophilic interactions, T-cadherin functions as a signaling receptor for two ligands—HMW (high molecular weight) adiponectin [2] and LDL (low density lipoproteins) [3]. An elevated level of LDL in the blood is associated with lipid accumulation in the vascular wall, contributing to atherogenesis and vessel dysfunction, whereas adiponectin exerts multiple protective effects in the cardiovascular system and metabolic tissues, including insulin sensitizing, anti-inflammatory, anti-proliferative, and anti-atherosclerotic actions [4,5]. Unlike other adipokines, adiponectin level is reduced in pathological conditions such as insulin resistance, type 2 diabetes mellitus (T2DM), obesity, metabolic syndrome, and cardiovascular diseases [1]. The competition of these two ligands for T-cadherin binding is potentially important in health and disease [6], suggesting that an imbalance of LDL and adiponectin in the blood might be a putative mechanism behind the pathological conditions [1].

Several papers have recently demonstrated that T-cadherin is essential for the adiponectin-mediated protective effects. T-cadherin and adiponectin co-localize in the cardiovascular system, specifically in aorta and heart tissue, as well as in skeletal muscles [7,8,9]. Using both cardiac hypertrophy and ischemia-reperfusion mice models, T-cadherin was shown to recruit the circulating adiponectin to the cardiac muscle. In T-cadherin-knockout mice, adiponectin failed to exert its protective effects and was not detected in the heart tissue despite its elevated plasma level [7]. In line with this, adiponectin was identified in skeletal muscles [8]. In the revascularization response to the chronic hind limb ischemia model, T-cadherin was shown to play a pivotal role. Both strains, T-cadherin-deficient and adiponectin-deficient mice, exhibited similar perturbed blood flow recovery phenotypes; however, the delivery of exogenous adiponectin rescued the impaired revascularization only in adiponectin-deficient mice, but not in mice devoid of T-cadherin [8].

Recently, it was demonstrated that the adiponectin/T-cadherin system enhances exosome biogenesis and secretion in T-cadherin–expressing endothelial cells in vitro and in the aorta in vivo, as well as exosomes in conditioned media and in the blood [10]. Upregulated exosome production in response to adiponectin was also displayed in the differentiated C2C12 myotubes [11] and adipose-derived mesenchymal stem cells (MSCs) [12]. Exosomes purified from the conditioned medium of human MSCs exhibited cardioprotective effects in a mouse model of ventricular hypertrophy caused by chronic heart failure. This effect was largely T-cadherin-dependent, since T-cadherin-knockout MSCs produced fewer exosomes in response to adiponectin in vitro and exhibited a substantially lower cardioprotective potential in vivo [12].

Obata et al. showed that the increased exosome production in endothelial cells was T-cadherin dependent and accompanied by the reduction of cellular ceramides through exosome-mediated ceramide efflux [10]. The elevated level of ceramide is thought to be one of the mechanisms of endothelial dysfunction in hypertension [13]. Moreover, one common single nucleotide polymorphism (SNP) located upstream of the *Cdh13* gene has been associated with hypertension in two independent GWAS studies conducted on populations from Germany and Estonia [14]. While numerous studies using experimental animal models (cited in [1]) suggest a conceivable role of T-cadherin in endothelial cell physiology, solid experimental evidence on the potential role of T-cadherin in hypertension is lacking. Therefore, we generated the Cdh13^∆Exon3^ mice model with truncated T-cadherin lacking exon 3 to study the phenotype of these mice, and thereby determined the role of T-cadherin in regulating blood pressure and physical stamina.

## 2. Results

### 2.1. T-Cadherin-Deficient Mice Genotyping Using PCR

To create T-cadherin-deficient mice we crossed Cdh13^loxP/loxP^ mice, possessing 2 loxP sites flanking exon 3 of the *Cdh13* gene, to mice constitutively expressing Cre-recombinase (Figure 1A). We isolated genomic DNA as described in the Materials and Methods section and analyzed the genotype of F2 pups using three primers (Figure 1A). We confirmed the absence of exon 3 in T-cadherin-deficient mice using the isolated total genomic DNA and PCR with specific primers (see Section 4). According to PCR product analysis, exon 3 was absent in knockout mice, since the 354bp product (primer #2 and primer #3) was not detected in the samples from T-cadherin-deficient mice (Cdh13^∆Exon3^ mice) (Figure 1B). These data verified the success of the Cre-mediated site-specific recombination reaction in having the annealing site for primer #2 precisely deleted (Figure 1A), in contrast to the 618 bp product detected in the samples of WT and heterozygous F2 pups (Figure 1B). Given that the presence of the 354 bp amplicon corresponded to the full-length T-cadherin in WT mice, the 618 bp amplicon, detected only in the samples from mice lacking the 3rd exon (knockout and heterozygous mice), clearly served as a control for exon 3 sequence elimination (Figure 1B).

To verify the frame shift in Cdh13^∆Exon3^ mice, the sequencing of *Cdh13* mRNA was performed (Figure 1C). The obtained results confirmed that exon 3 was missing in Cdh13^∆Exon3^ mice, with a presumptive stop codon being formed at the place of its deletion and transcript formation encoding only the leading peptide with the length of 52 amino acids (Figure 1C).

Further, to confirm the absence of exon 3 in the *Cdh13* gene in F2 generation and separate WT, heterozygous, and Cdh13^∆Exon3^ mice, we applied real-time PCR using TaqMan probes. The F2 generation comprised 63 mice. The test system included four primers complementary to the *Cdh13* gene and two TaqMan probes. The first primer pair carried the FAM fluorophore and was specific to the 2nd exon in the *Cdh13* gene, while the second primer pair carried the HEX fluorophore and was specific to the 3rd exon. Amplification was detected in two channels: FAM and HEX. The samples with genomic DNA being amplified in both channels were considered to be the wild-type mice, while the DNA samples being amplified only in the FAM channel, but not in the HEX channel, corresponded to T-cadherin-deficient mice lacking the 3rd exon. Amplification of the 2nd exon was detected in all 63 examined mice, while the 3rd exon was detected only in 16 mice, indicating that 47 mice in F2 generation were T-cadherin-deficient (Cdh13^∆Exon3^ mice); these mice were subsequently used for breeding. The *Plaur* gene (reference gene) was amplified in all mice tested, confirming the specificity of the results. The amplification cycles of the products corresponding to the 2nd and 3rd exons in the *Cdh13* gene are presented in Appendix A.

We also applied qPCR to verify the *Cdh13* gene modification using murine samples from the aorta, brain, heart, muscle, and kidneys, where T-cadherin is highly expressed [15,16]. The samples of aorta, brain, and muscle tissues were obtained from three WT and three Cdh13^∆Exon3^ mice, the heart samples were obtained from three WT and two Cdh13^∆Exon3^ mice, while one kidney sample was obtained from one Cdh13^∆Exon3^ mouse. In all samples from Cdh13^∆Exon3^ mice T-cadherin expression was not detected (Cq > 40), except in one aorta sample from a Cdh13^∆Exon3^ mouse, where the expression level was 0.00102 ± 0.00004 (mean ± SEM) of the WT level. Therefore, the obtained results clearly indicated the presence of mRNA, containing the 3rd exon, only in WT mice.

### 2.2. T-Cadherin Expression Analysis in WT and T-Cadherin-Deficient Mice Using Western Blot

To verify that the introduced modification led to the suppressed expression of T-cadherin protein, we performed Western blot analysis using heart, muscle, and aorta tissues from Cdh13^∆Exon3^ and WT mice. For that, we used the commercially available antibodies raised against the N-terminus peptide of T-cadherin corresponding to the exon 3 region (Abcam #167407, target peptide—aa 100–200). Both 130-kDa pro-protein and proteolytically cleaved 105-kDa, a mature form of T-cadherin, were readily detected in the tissues of WT mice, but not in those of Cdh13^∆Exon3^ mice (Figure 1D). We confirmed the absence of exon 3 in Cdh13^∆Exon3^ mice with a plausible stop codon being formed at the place of its deletion (Figure 1C); however, the residual T-cadherin was detected in the membrane fraction of Cdh13^∆Exon3^ mice (Figure 2A), suggesting the presence of truncated T-cadherin, which can be synthesized from the downstream start codon in exon 6 (Figure 2B).

### 2.3. Phenotypic Analysis and Resting Blood Pressure Evaluation in T-Cadherin-Deficient Mice

The obtained Cdh13^∆Exon3^ mice exhibited normal gross morphology, life expectancy, and breeding capacity. However, in our model, the body weight of Cdh13^∆Exon3^ mice was significantly lower compared to the wild type (weight median 20.95 g interquartile range (19.7; 21.8), N = 12 vs. 22.4 g (22; 24.2), N = 10, *p* = 0.042) (Figure 3). We have measured resting blood pressure in four wild-type and six Cdh13^∆Exon3^ mice preliminarily familiarized with the procedure. Median (interquartile range) systolic blood pressure was 70 (61; 85) mm Hg in wild-type mice (N = 10) and 99 (84; 105) in knockout mice (N = 10) *p* = 0.1714. Median (interquartile range) diastolic blood pressure was 40 (32; 60) mm Hg in wild-type mice and 56 (50; 69) in knockout mice (*p* = 0.2571). Median (interquartile range) heart rate was 616 (483; 641.5) mm Hg in wild-type mice and 456 (396; 551) in knockout mice (*p* = 0.1714). Therefore, resting blood pressure in Cdh13^∆Exon3^ mice was moderately higher, however the difference was not significant.

### 2.4. T-Cadherin Deficiency Is Associated with Increased Blood Pressure

Despite the lack of statistical significance, the observed tendency towards elevated resting blood pressure allowed us to hypothesize that the cardiovascular system of Cdh13^∆Exon3^ mice responds differently to stress. To test this hypothesis, we compared the ability of wild type and Cdh13^∆Exon3^ mice to run on a treadmill for a sustained period of time. We found that the time spent running and the distance covered by both Cdh13^∆Exon3^ and WT mice did not differ. However, systolic blood pressure was higher in Cdh13^∆Exon3^ mice compared to WT mice (median 92 mmHg, interquartile range (77; 97), N = 8 vs. 74 mmHg, interquartile range (59; 80), N = 11) (Figure 3). These data suggest that the absence of full-size T-cadherin prevents the cardiovascular system from adapting to the stress inflicted by intensive physical activity.

### 2.5. T-Cadherin Deficiency Affects Adiponectin Level and AMPK-Dependent Signaling

Since T-cadherin expression is closely tied to the expression of its ligand-adiponectin, we assumed that T-cadherin deficiency can affect the adiponectin plasma level. We compared the plasma adiponectin levels of WT and Cdh13^∆Exon3^ mice using an ELISA kit (SEA 605Mu, Cloud-Clone, Wuhan, China). Indeed, the adiponectin plasma concentration was 4.37-fold higher in Cdh13^∆Exon3^ mice compared to WT (0.27 ± 0.05 ug/mL vs. 1.18 ± 0.18; *p* = 0.008). However, despite the observed decline tendency in adiponectin concentration in myocardium, no significant difference was detected (25 ± 2.3 ng/mg in Cdh13^∆Exon3^ mice vs. 33.6 ± 8 in WT mice, *p* = 0.05533).

Among the signaling pathways triggered by adiponectin, the AMPK signaling axis is the most important for adiponectin-mediated effects on skeletal muscles and the cardiovascular system [17,18]. Therefore, we analyzed the basal level of AMPK phosphorylation and detected no significant difference between Cdh13^∆Exon3^ and WT mice. However, after running on the treadmill, the AMPK phosphorylation level was higher in the skeletal muscles and myocardium of Cdh13^∆Exon3^ mice compared to WT mice (Figure 4).

## 3. Discussion

T-cadherin has been recognized as an exclusive receptor for HMW adiponectin, the most metabolically active form of adiponectin [2,19]. Native adiponectin specifically interacts with T-cadherin in the blood, but not with the other adiponectin receptors, AdipoR1 and AdipoR2 [20]. Moreover, T-cadherin recruits adiponectin to organs and tissues, ensuring that adiponectin’s protective functions will benefit them [7,8,9,21]. In T-cadherin-KO mice, adiponectin fails to associate with skeletal muscles and myocardium despite the elevated adiponectin concentration in circulation [7].

The pertinent literature indicates that the adiponectin–T-cadherin axis plays an essential role in vascular protection, while T-cadherin deficiency intensifies endothelial dysfunction and delays NO production [21,22,23]—the conditions associated with hypertension. In humans, a lower level of adiponectin is linked to elevated blood pressure as compared to normotensive adults, while augmented adiponectin in the blood is associated with a reduced risk of hypertension [24]. Meanwhile, the question about the exact mechanisms of adiponectin action and the potential receptors involved in regulation of blood pressure remains to be resolved.

Since comprehensive literature data on the role of T-cadherin in hypertension is missing, we generated Cdh13^∆Exon3^ mice and examined their physical stamina and blood pressure. First, we confirmed the absence of exon 3 in Cdh13^∆Exon3^ mice (Figure 1). Unexpectedly, subcellular fractionation followed by WB analysis using antibodies against the C-terminus of T-cadherin demonstrated the presence of residual T-cadherin in the membrane fraction in Cdh13^∆Exon3^ mice (Figure 2A). Although this assumption requires further exploration, the rationale behind the presence of truncated T-cadherin in Cdh13^∆Exon3^ mice, albeit in a low quantity, is that T-cadherin can be synthesized from the downstream start codon in exon 6 in Cdh13^∆Exon3^ mice (Figure 2B). The resulting molecular weight of these bands was quite similar to the bands of WT mice (Figure 2A). This could be explained by either of two processes: the cleavage of the signal peptide and the pre-peptide resulting in the mature protein [1], or the known extensive glycosylation of T-cadherin [25], which overall levels out the difference in molecular weight.

The obtained Cdh13^∆Exon3^ mice displayed the overall normal gross morphology, life expectancy, and breeding capacity. Yet the body weight of T-cadherin-deficient mice was significantly lower than that of WT mice (Figure 3), implicating the importance of this receptor for the developmental program. The time spent running and the distance covered during a treadmill test were the same for Cdh13^Exon3^ and WT mice. While the resting blood pressure in Cdh13^∆Exon3^ mice was moderately higher than in the control, the intensive physical training led to a notable rise in systolic blood pressure (Figure 3).

Our data are in accord with the only currently published paper addressing the role of T-cadherin in blood pressure regulation, where it was demonstrated that SNP, located upstream of the *Cdh13* gene and potentially altering T-cadherin expression and/or function, was associated with hypertension in European populations [14]. In the present study, we demonstrate that the full-length T-cadherin is required for cardiovascular adaptation to stress triggered by intensive physical exercise.

Recent data suggest that adiponectin interaction with T-cadherin facilitates the production of exosomes and, therefore, accelerates the exchange rate of the plasma membrane in vascular cells [10]. Dynamics of the plasma membrane exchange may be a critical step in the regulation of vascular cells’ response to hormones, such as adiponectin, which can lower the blood pressure and be produced in response to stress mediated by intensive physical activity. Further studies are expected to determine the underlying molecular mechanisms of T-cadherin–adiponectin mediated effects on the regulation of body weight and blood pressure.

Emerging evidence indicates that the *Cdh13* gene expression and the adiponectin plasma level are intricately intertwined [1]. A GWAS study demonstrated the presence of reduced HMW adiponectin in the blood serum of patients with SNP rs3865188 in the *Cdh13* gene [26]. A rare promoter methylation in the *Cdh13* gene (SNPs rs8060301 and cg09415485) demonstrated a strong association and a 4.5-fold decrease in plasma adiponectin levels [27]. A prime example of T-cadherin and adiponectin interrelationships has been presented by Parker-Duffen et al. using experimental animal models [8]. Despite a remarkably high level of plasma adiponectin in T-cadherin-KO mice, adiponectin was unable to bind the heart tissue and activate AMPK signaling, pointing to a critically important role of T-cadherin [8]. In the search for the possible mechanism behind the observed effects of T-cadherin deficiency linking it to adiponectin signaling, we evaluated the adiponectin levels in blood and heart tissue. Compared to WT mice myocardium, only a downward tendency in adiponectin content was detected in Cdh13^∆Exon3^ mice. In line with the previously published results [7,28], we found a significant increase in the level of circulating adiponectin in Cdh13^∆Exon3^ mice: a 4.37-fold rise in Cdh13^∆Exon3^ mice was detected compared to WT mice. Therefore, our data suggest that it is the efficiency of adiponectin interaction with the full-length T-cadherin that is particularly important for blood pressure control, rather than the level of this circulating hormone per se.

Among the signaling pathways triggered by adiponectin, the AMPK axis is the most prominent for adiponectin-mediated effects in skeletal muscles and the cardiovascular system [17,18]. Unexpectedly, our data revealed that intensive physical training leads to an elevated level of AMPK phosphorylation in the skeletal muscles and myocardium of Cdh13^∆Exon3^ mice, despite the decreased tissue adiponectin content and the lack of full-length T-cadherin expression (Figure 4). One of the tentative explanations for these results is that in these conditions adiponectin may interact with the other adiponectin receptor, AdipoR1, triggering compensatory intracellular signaling events, including AMPK pathways [29]. These data call for a more detailed investigation, as they raise questions more than they provide answers.

To infer, our data shed light on a previously dismissed role of T-cadherin in the regulation of blood pressure and stamina in mice, contributing to our understanding of the pathogenesis of hypertension.

## 4. Materials and Methods

### 4.1. Animals

T-cadherin-KO mice were generated by crossing *Cdh13^floxedfloxed^* mice to mice with X-linked constitutive expression of Cre-recombinase, both on C57Bl/6N background (see Section 2 for the breeding scheme). *Cdh13^floxedfloxed^* mice were a kind gift from Prof. Gerard Karsenty, Columbia University, USA [30], while mice with X-linked constitutive expression of Cre-recombinase (B6.C-Tg (CMV-cre)1Cgn/J, Stock # 006054) were obtained from Jackson Laboratory. All animals were kept in the Animal Core Facility of the Faculty of Medicine, Lomonosov Moscow State University, on a regular 12 h light/12 h dark cycle in a standard controlled environment. Food and water were provided ad libitum. All procedures with live animals including housing, handling, experimental manipulations, and euthanasia were conducted according to Directive 2010/63/EU and approved by the bioethical commission of Lomonosov MSU (#3.5 from 17 March 2022). We used male mice aged 8 to 10 weeks in all experiments. Blood pressure in mice was measured non-invasively using a system involving tail-cuff and tail-bloodstream sensors (Neurobiotics, Moscow, Russia). During measurements, mice were placed in restrainers on a heated platform. To train mice for the procedure (treadmill test), the measurements were performed daily for 5–7 days until the values became reproducible. After that, blood pressure was measured for an additional 4 days and mean values were calculated. Blood was collected from hearts into a heparin buffer. Hearts were then perfused with phosphate-buffered saline (PBS) to wash off the blood prior to protein and RNA extraction. Hind limb muscle was used as a skeletal muscle with high exercise stress response. Tissues were immediately frozen in liquid nitrogen for further analysis.

### 4.2. Treadmill Test

The stamina of Cdh13^∆Exon3^ and WT mice was assessed with the treadmill test as described by Dougherty and co-authors [31]. Mice were pre-trained before the treadmill test. On the first day, they were placed on individual tracks for 3 min with the electric grid on, then the track was run at 1.5 m/min to ensure that mice understood the task and proceeded to run. Later the speed was raised to 8 m/min following the scheme: 5 min–9 m/min, 7 min–10 m/min, 10 min–the end of training. The next day mice were trained according to the scheme: 0 min–10 m/min, 5 min–11 m/min, 10 min–12 m/min, 15 min–the end of training. The third day of training was similar to the second day. The day before the experiment, the mice were not trained.

Measurement of mice endurance began by placing the mice on separate treadmills with electric grids turned on, and a treadmill speed of 12 m/min was set. The speed was then increased using the following scheme: 0.5 min–14 m/min, 1 min–16 m/min, 6 min–18 m/min, 30 min–20 m/min, 45 min–22 m/min, 60 min–24 m/min, 75 min–26 m/min. The mice were performing the task until they were in a zone equal to one mouse body length from the electric grid for 5 s in a row. After that, the experiment was terminated and the results (distance, m and time, min) were recorded.

### 4.3. PCR-Based Genotyping of Mice

Genomic DNA from ear biopsies after proteinase K digestion was used for mouse genotyping. We designed three primers: two forward (5′-ACTGAGGCATTCAAGTTCGGT-3′ and 5′-TTTCCCCATCAACTGGCACA-3′) and one reverse primer (5′-GCAGGGTTGTGCATCACTAGA-3′). This system allowed us to identify WT and knockout alleles in any combination (Figure 1). We used 10 cycles of touchdown PCR (95 °C—15 s, 60 °C (−0.5 °C/cycle)—10 s, 72 °C—40 s) followed by 30 cycles of standard PCR (95 °C—15 s, 55 °C—10 s, 72 °C—40 s).

### 4.4. cDNA Sequencing

For the partial sequencing of T-cadherin mRNA in knockout mice, total RNA was isolated from brain samples of knockout mice with the RNeasy Mini Kit (Qiagen, Venlo, The Netherlands). Reverse transcription was performed with a MMLV RT kit (Evrogen, Moscow, Russia) to amplify T-cadherin cDNA fragments with the following primers: forward ATGCAGCCGAGAACTCCGCT, reverse GTCATCGATCACGATGGTGGCTGT. The resulted PCR product was cloned into the pAL-T2 rapid ligation vector (Evrogen, Moscow, Russia) using T4 ligase (Sibenzyme, Novosibirsk, Russia). The insert was sequenced using T7dir and M13rev oligonucleotide primers (Evrogen, Moscow, Russia).

### 4.5. Quantitative PCR (qPCR) Genotyping of Mice

To evaluate T-cadherin expression in WT and Cdh13^∆Exon3^ mice, qPCR was performed. Tissue samples were flash-frozen in liquid nitrogen immediately after isolation. RNA was isolated using the RNeasy Mini Kit (Qiagen, Venlo, The Netherlands). We carried out reverse transcription using a MMLV RT kit (Evrogen, Moscow, Russia). To amplify the *Cdh13* gene, the following primers placed in exons 3 and 4 were used: forward 5′-CATGGCAGAACTCGTGATTG-3′, reverse 5′-GGTTCTCTGGGATCAAGATGG-3′. *Rplp0* was used as a reference gene with the following primers: forward 5′-GAGGAATCAGATGAGGATATGGGA-3′, reverse 5′-AAGCAGGCTGACTTGGTTGC-3′. PCR was performed using qPCRmix-HS (Evrogen, Moscow, Russia), and a CFX96 PCR machine (Bio-Rad, Hercules, CA, USA).

### 4.6. RT-PCR for Genotyping of Mice

For mice genotyping and subsequent selective breeding, blood sampling was conducted in 1 mL disposable tubes. Next, the blood was subjected to erythrocyte washing. Genomic murine DNA was then isolated from the precipitated leukocytes. DNA isolation was performed using the “Proba-NK” kit (“DNA-Technology”, Moscow, Russia) according to the manufacturer’s instructions. The isolated DNA, in a volume of 100 μL, was used for RT-PCR with Taq-Man fluorescent probes. The primer pairs and corresponding probes are presented in Table 1. The *Plaur* gene (the urokinase receptor) was used as a reference gene for the isolated murine genomic DNA. Amplification was carried out in a volume of 35 μL using the following program: 50 cycles of 94 °C—10 s, 68 °C—20 s, 72 °C—10 s. The fluorescence level was measured in each cycle using two channels (FAM and HEX) at a temperature of 68 °C.

### 4.7. Western Blotting

Proteins were extracted from hearts and hind limb muscles by homogenization in liquid nitrogen, followed by solving in Laemmli buffer (Bio-Rad, Hercules, CA, USA) containing protease and phosphatase inhibitors, as well as beta-mercaptoethanol (Sigma-Aldrich, St. Louis, MO, USA). Samples were subjected to SDS-PAGE next to the protein ladder (Spectra Multicolor High Range Protein Ladder, 26,625 or 26,619, ThermoFisher, Waltham, MA, USA), followed by the transfer to PVDF membrane. Membranes were blocked in 5% skim milk for 12 h at +4 °C and probed with rabbit primary antibodies anti-AMPK alpha, (1:500, CellSignaling, Danvers, MA, USA, #2532), anti-phospho-AMPK alpha (1:500, Affinity Biosciences, Cincinnati, OH, USA, AF3423), anti-T-cad (1:1000, Abcam, Cambridge, UK, ab167407), and anti-beta-actin (1:5000, Cloud-Clone Corp., Wuhan, China, PAB340Mi01), followed by incubation with horseradish peroxidase-conjugated secondary antibodies (1:3000, IMTEK, Moscow, Russia). Protein bands were visualized using enhanced chemiluminescence with Clarity Max Western ECL Blotting Substrate (Bio-Rad) registered on a ChemiDoc instrument (Bio-Rad, Hercules, CA, USA) according to the manufacturer’s protocols. Image analysis was performed in Image Lab (Bio-Rad).

To separate the cytoplasmic fraction and the fraction enriched with membrane proteins, we used the Pierce Subcellular Fractionation kit for tissues (Thermo Fischer Scientific, Waltham, MA, USA), performing all manipulations at +4 °C. The hearts of 8-week-old WT and Cdh13^∆Exon3^ knockout mice were isolated, minced and thoroughly homogenized in 10× (V/m) volume of Cytosolic extraction buffer containing protease inhibitors, followed by a 10 min incubation on ice and filtration through the mesh supplied by the manufacturer. Filtrates were centrifuged at 500× *g* for 5 min, resulting in supernatant (cytosolic fraction) and pellet (fraction enriched in membrane proteins) further used for sample preparation and electrophoresis.

Samples were subjected to SDS-PAGE, followed by the transfer onto Nitrocellulose membrane (Thermo Scientific 88018) next to the protein ladder (Spectra Multicolor High Range Protein Ladder, 26,625 or 26,619, ThermoFisher). Membranes were blocked in 5% skim milk for 12 h at +4 °C and were probed with the primary antibodies anti-T-cadherin (rabbit polyclonal, 1:500, sc-7940, Santa Cruz Biotechnology, Dallas, TX, USA) and anti-alpha tubulin (mouse monoclonal, clone DM1A, 1:50,000, Sigma–Aldrich, MABT205), followed by incubation with horseradish peroxidase-conjugated secondary antibodies (1:8000, Jackson ImmunoResearch, West Grove, PA, USA), donkey anti-mouse HRP (#715-035-151), donkey anti-rabbit (#111-035-003). Protein bands were visualized using enhanced chemiluminescence with the Affinity ECL kit (Affinity, KF003) and registered on a ChemiDoc instrument (Bio-Rad, Hercules, CA, USA) according to the manufacturer’s protocols. Image analysis was performed using the Fiji program (NIH, Bethesda, MD, USA).

### 4.8. ELISA of Adiponectin

Adiponectin concentration was measured in heparin plasma and heart homogenates using an ELISA kit (SEA 605Mu, Cloud-Clone, Wuhan, China). Hearts were placed on ice, then washed by ice-cold PBS; 100 mg of each heart was homogenized in 3 mL of lysis buffer (50 mM HEPES, 0.2% Triton X-100, 1 mM EDTA, 0.1 mM PMSF) using gentle MACS Dissociator (Miltenyi Biotec, Bergisch Gladbach, Germany) in M-tubes (running mouse heart 2.2. protocol). Homogenates were centrifuged (15 min at 10,000× *g*) and snap frozen in liquid N2. 100 mL from homogenate or plasma samples diluted by 1:2000 was applied in each well and processed further according to the manufacturer’s protocol.

### 4.9. Statistical Analysis

Continuous variables were analyzed with the Mann–Whitney U test. For multiple comparisons, the Kruskal–Wallis test was used. Statistical analysis was performed using Statistica 12.0 64 Software (StatSoft, Inc., St. Tulsa, OK, USA).

## Figures and Tables

**Figure 1 ijms-24-14204-f001:**
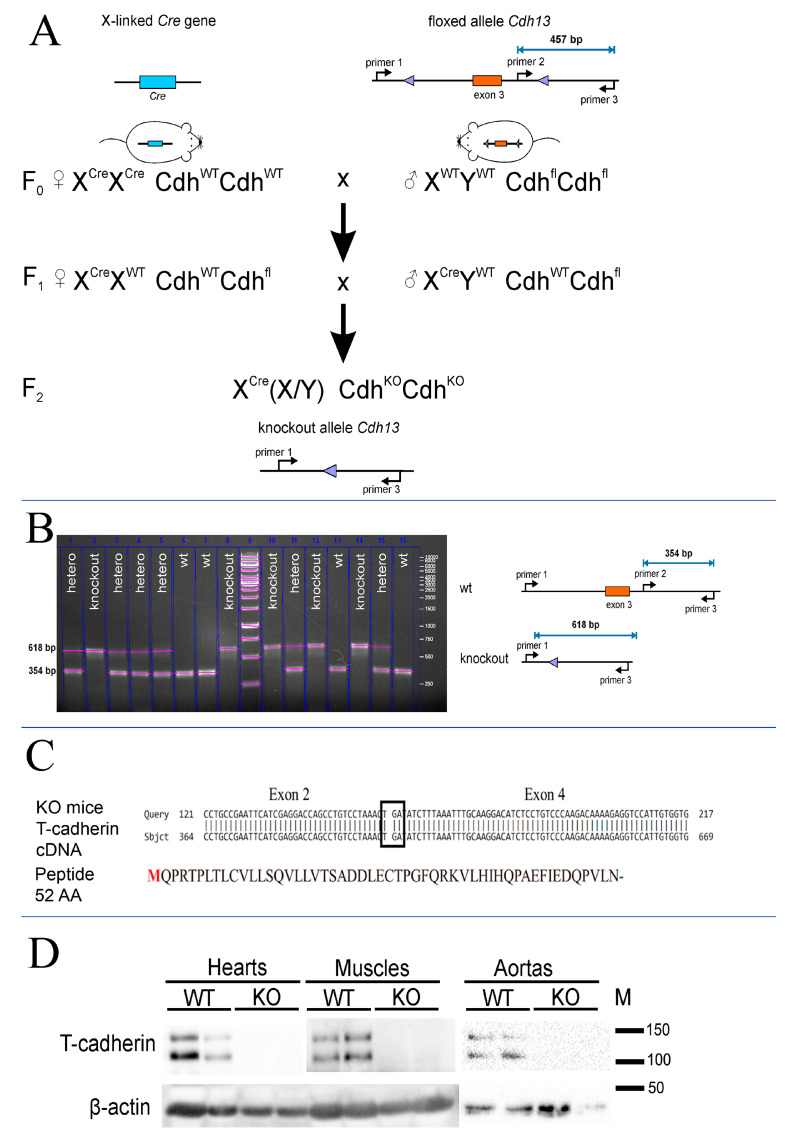
Cdh13^Exon3^ mice generation. (**A**) Breeding scheme to generate Cdh13^∆Exon3^ mice. (**B**) PCR product analysis and electrophoresis; a scheme for evaluating *Cdh13* gene expression in F2 offspring. The presence of the 354 bp amplicon corresponds to the full-length T-cadherin in WT mice. The 618 bp amplicon has been detected only in mice lacking the 3rd exon (knockout and heterozygous mice) due to Cre recombination. (**C**) Results of the T-cadherin cDNA sequencing from Cdh13^∆Exon3^ mice. The cDNA of T-cadherin from knockout mice was generated using the T-cadherin specific primers targeted to the N-terminus of T-cadherin mRNA. The cDNA sequencing confirmed the direct link of the 2nd and 4th exons with the appearance of a stop codon (TGA) and termination of the reading frame. The expected T-cadherin peptide results from an open reading frame of T-cadherin mRNA starting from the conventional start codon and corresponds to 52 amino acids. (**D**) T-cadherin content in the hearts, skeletal muscle, and aortas of wild type (WT) and Cdh13^∆Exon3^ (KO) mice was analyzed using antibodies against the synthetic peptide located in the exon 3 region (Abcam #167407, target peptide—aa 100–200). Cdh13^∆Exon3^ (KO) mice lack the full-length T-cadherin.

**Figure 2 ijms-24-14204-f002:**
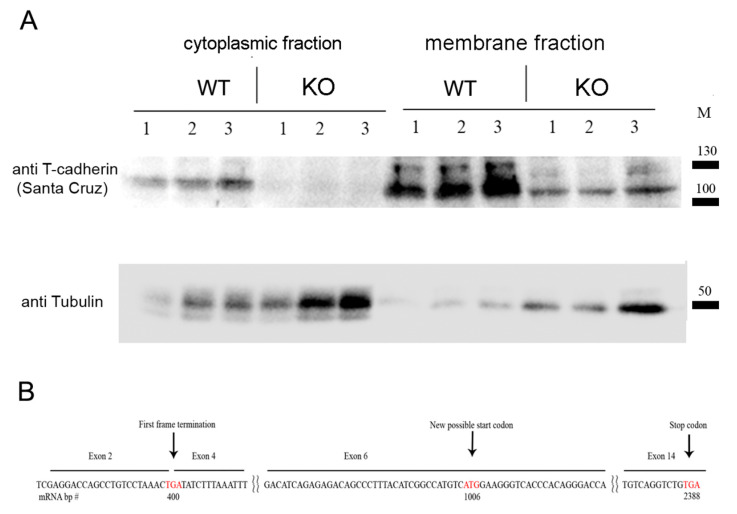
(**A**) Western blot analysis of cytoplasmic and membrane enriched fractions isolated from mouse hearts of WT and Cdh13^∆Exon3^ (KO) mice (parallel triple sampling for each mouse type). Commercially available antibodies raised against the C-terminus of T-cadherin revealed the residual T-cadherin signal, albeit in a lower amount, in the membrane fraction in Cdh13^∆Exon3^ mice compared to WT samples. For loading control, anti-alpha tubulin antibody was used. (**B**) mRNA (cDNA) scheme demonstrates an alternative start codon position for truncated T-cadherin translation in Cdh13^∆Exon3^ (KO) mice. Stop and start codons are labeled by the red font.

**Figure 3 ijms-24-14204-f003:**
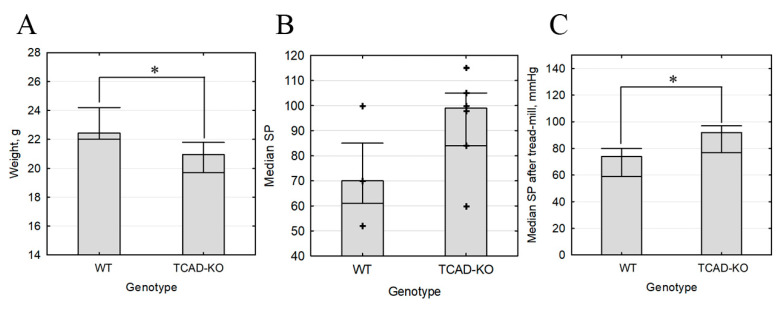
Body weight and blood pressure of Cdh13^∆Exon3^ (KO) vs. wild type (WT) mice. (**A**) Body weight (median, N = 10, * *p* = 0.042). (**B**) Resting systolic blood pressure (N = 10, *p* > 0.05). (**C**) Stress systolic blood pressure (N = 10, * *p* = 0.046).

**Figure 4 ijms-24-14204-f004:**
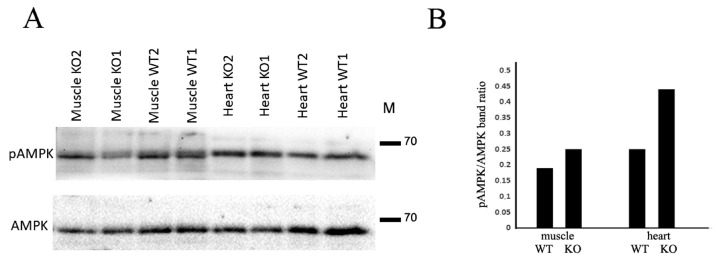
T-cadherin deficiency affects AMPK phosphorylation. (**A**) Representative immunoblots of skeletal muscle and heart tissues from WT and Cdh13^∆Exon3^ (KO) mice. (**B**) Densitometry of immunoblot in (**A**) presented as a ratio of pAMPK to total AMPK.

**Table 1 ijms-24-14204-t001:** Primer sequences used for mouse genotyping.

Primers	Sequence	Gene	Polymerization Direction
Prd-s1	-CTGGATTCCAGCGGAAAGTGT-	*Cdh13*	→
Ptd1	(BHQ1)-CATCCACCAGCCTGCCGAATTCA-(FAM)	*Cdh13*	*TaqMan* Probes
Prd-a1	-AGTTTAGGACAGGCTGGTCCT-	*Cdh13*	←
Pri-s1	-ACAGCGACGGCACCTTAGTGG	*Cdh13*	→
Pti1	(BHQ1)-TGCGGTGGGCAGGACCCTGTTTGT-(SIMA)	*Cdh13*	*TaqMan* Probes
Pri-a1	-TGGATGTCTTTGCCCCCGACA-	*Cdh13*	←
Plr1fs	-GACCCACCTCAACGTCTCTGT-	*Plaur*	→
Plrfq1n	(BHQ1)-CTCCCTCCCCCCTGCTCCATCT-(FAM)	*Plaur*	*TaqMan* Probes
Plrf1a	-CCCGCAGCCCTTGTTCCAAT-	*Plaur*	←

## Data Availability

All relevant data are included within the paper and its additional file.

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
