# Peer review of "T-Cadherin Deficiency Is Associated with Increased Blood Pressure after Physical Activity"

_ijms, 2023, doi:10.3390/ijms241814204_

Round 1
Reviewer 1 Report
This paper is significant in that it shows the importance of the relationship between T-cadherin and adiponectin in blood pressure control under exercise stress and its mechanism. However, the following points should be discussed and the rationale for reaching conclusions should be provided.
1. p6 l186: Authors described that “systolic blood pressure was higher in Cdh13∆Exon3 mice comparing to WT mice (median 92 mmHg, interquartile range [77;97], N=8 vs. 74 mmHg, interquartile range [59;80], N=11) (Figure 3)”
-> Authors should have kept treadmill-trained mice for a period of time. If the authors are investigating the lifespan of Cdh13∆Exon3 and WT mice, authors should show the results in this manuscript.
2. p7 Figure 3: 1) There are no A, B and C marks in the three figures, authors should add them.
2) As for the resting systolic blood pressure in Fig. 2, the difference in the average values is large, so the resting systolic blood pressure for each individual should be shown as a point to show the trend of variance.
3. p7 l239: Authors should draw Kaplan-Meier survival curves on Cdh13∆Exon3 and WT mice.
4. Rubina et al [1] described the importance of Adipose tissue for the effects of T-cadherin. Adipose tissue should also be examined in this study.
5. The discussion should also describe the relationship between AMPK and brown adipocytes.
6. p8l285: Authors should also refer to Zhu et al. 2022 below.
Zhu, Di, et al. "Targeting adiponectin receptor 1 phosphorylation against ischemic heart failure." Circulation Research 131.2 (2022): e34-e50.
Reviewer 2 Report
The authors aimed to elucidate the role of T-cadherin, which binds to the anti-atherogenic factor HMW adiponectin and directs it to myocardium, arterial walls and skeletal muscle, in atherogenesis and hypertension. They engineered mice deficient in exon 3 of T-cadherin, which has been implicated in its cardiovascular effects, in the target tissues to determine the factor's role in atherosclerosis-associated hypertension. There are serious questions about whether the experimental approach was sufficient to accomplish the aims of the study. First, the systolic blood pressures they measured in control mice were low (70 mm Hg), which is > 90 mm Hg in previous studies. The heart rates may also be a bit high (616). There is no indication that methods of measurement were validated, for instance by employment of alternate methods. The results presented, which generally show marginal differences between control and KO animals despite their statistical significance, also leave some doubt about whether T-cadherin function was really knocked out. The investigators should have assessed cytosolic and membrane markers to verify that they isolated the pure fractions. Other issues: Why were only male mice used? Reference formats for both in-text citations and references seem irregular. All-in-all, the authors need to reassess the validity of their methods.
Generally, the English usage is satisfactory, but there are places where it breaks down (e.g., lines 196-197, 205, 233, 238, 274-277, 282).
Round 2
Reviewer 2 Report
Apparently, the authors contend that this submission should be considered a largely preliminary study indicating future research directions. Within this context, they have made whatever suggested changes that they could.
There are still two instances (lines 196 and 205) of "comparing to" instead of "compared to."